# Adaptive Nonlinear Control of Salient-Pole PMSM for Hybrid Electric Vehicle Applications: Theory and Experiments

Chaimae El Fakir [1,†], Zakariae El Idrissi [2,*,†], Abdellah Lassioui [2], Fatima Zahra Belhaj [2], Khawla Gaouzi [2], Hassan El Fadil [2] and Aziz Rachid [2,3]

1 IMS Laboratory—UMR CNRS 5218, University of Bordeaux, 351 Cours de la Libération, 33405 Talence, France
2 ISA Laboratory, ENSA, Ibn Tofail University, Kenitra 14000, Morocco
3 LSIB Laboratory, FST, Hassan II University of Casablanca, Mohammedia 28806, Morocco
* Correspondence: zakariae.elidrissi@uit.ac.ma; Tel.: +212-640-356-189
† These authors contributed equally to this work.

**Abstract:** This research work deals with the problem of controlling a salient-pole permanent-magnet synchronous motor (SP-PMSM) used in hybrid electric vehicles. An adaptive nonlinear controller based on the backstepping technique is developed to meet the following requirements: control of the reference vehicle speed in the presence of load variation and changes in the internal motor parameters while keeping the reliability and stability of the vehicle. The complexity of the control problem lies on the system nonlinearity, instability and the problem of inaccessibility to measure all the internal parameters, such as inertia, friction and load variation. For this issue, an adaptive backstepping regulator is developed to estimate these parameters. On the basis of formal analysis and simulation, as well as test results, it is clearly shown that the designed controller achieves all the goals, namely robustness and reliability of the controller, stability of the system and speed control, considering the uncertainty parameters' measurements.

**Keywords:** salient-pole permanent-magnet synchronous motor (SP-PMSM); hybrid electric vehicle (HEV); adaptive nonlinear controller; backstepping control





## 1. Introduction

Robust and fast processes are required to support market developments, and this remains valid for electric motor design. Permanent-magnet synchronous motors (PMSMs) are used in different fields of high-performance electric drive. They have many advantages, such as high-power density, high efficiency, high torque/mass, high power/mass ratios, zero losses due to the rotor Joule effect, high reliability, and better controllability. In fact, this type of machine is recognized by its advantages as reported in [1–6]. However, SP-PMSMs have the power to offer a smaller size for more compact mechanical assemblies, as it is the case for electric cars [7,8].

This work was part of a hybrid electric vehicle project [9–11], and the choice of this type of machine was no accident. Additionally, the salient-pole permanent-magnet synchronous machine seems to be the best choice for a hybrid electric vehicle, because the total average torque produced by this machine is the sum of the synchronous and reluctant torques [12–14]. Furthermore, synchronous permanent-magnet motors tend to have a very high initial cost compared to other types of motors due to high magnet prices, but their prices are falling as technology develops. Recently, the high-performance SP-PMSM drive system has been developed, which has positively impacted their reputation in electric vehicle applications. Nonlinear control techniques of the SP-PMSM have been widely discussed in many types of research [15–17].

Nowadays, the problem of controlling uncertain systems is an attractive subject since vehicles require precision. Consequently, there have been many researchers interested in the nonlinear control problem in addition to the nonlinear adaptive control [18,19].

The nonlinear adaptive controller was presented in [15] based on a high-gain observer (HGO-NAC) of the SP-PMSM to control the mechanical rotational speed to its desired reference, regardless of the uncertainties of the system parameters and the unknown disturbance of the rapidly varying load torque. The presented results achieve the desired objectives but exhibit an overshoot of 5% over the reference speed at the moment of torque change, and the response time is approximately 1 s. Other authors used a non-linear state observer to control the rotor speed of the PS-PMSM [16]. The stability of the extended Luenberger observer is guaranteed by Lyapunov; hence, the results of this work satisfy the objectives of the elaborated theoretical analysis.

The sliding-mode observer-based design method for SP-PMSM backstepping control was proposed in [17]. The stability of this "controller-observer" approach is studied by the Lyapunov theorem. The presented results show a significant overshoot of speed and load torque, and a ripple at current $i_d$, which may degrade the machine performance. Another adaptive model reference system technique for speed estimation is used in [18] for the sensorless speed control of SP-PMSMs with space-vector pulse-width modulation. The stable and efficient estimation of the rotor speed is guaranteed by the simultaneous identification of the parameters of the SP-PMSM.

However, there are also studies on the speed-control problem of SP-PMSM drives based on the backstepping control and the nonlinear disturbance observer [19]. On the other hand, our paper deals with the speed-control problem by using a backstepping adaptive control method, which is a different approach.

In the present article, the focus is made on SP-PMSMs with unknown load torque. A typical nonlinear system is illustrated in Figure 1. The objective of the control is to closely regulate the SP-PMSM speed with load variations are present. Moreover, this speed should be regulated to the desired value despite the internal SP-PMSM parameters, which are considered unknown and constantly changing. The second objective is to estimate these parameters and the load torque simultaneously through the proposed adaptive backstepping controller. This adaptive regulator design is then performed to achieve closed-loop stability, tight output regulation, fast transient response and a good estimation of system internal parameters and load variations.

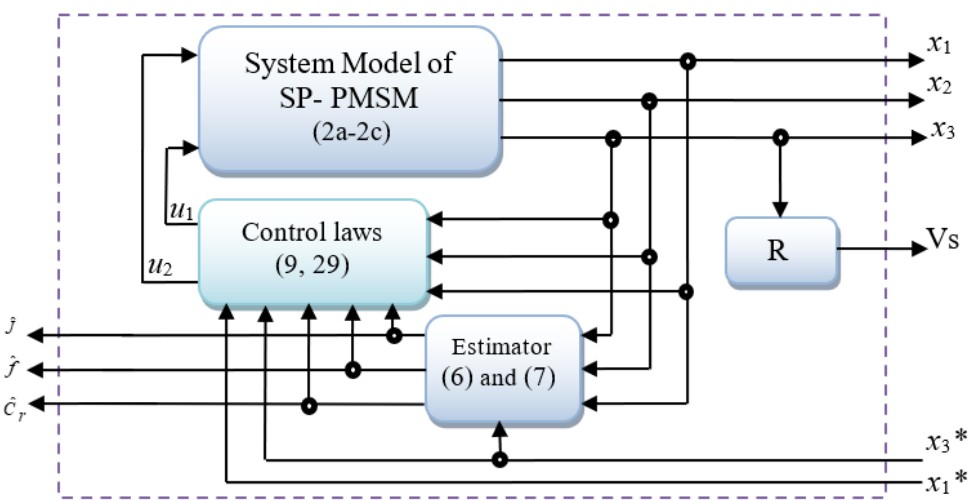

**Figure 1.** The SP-PMSM and control-system schematic diagram. Where $V_s = R\,x_3$: Vehicle Speed (m/s); R = 0.29 m: Radius of the point's trajectory (m).

The strength of our controller lies in the estimation of the unmeasured parameters of the SP-PMSM. Simulations with MATLAB®/Simulink® and experimental tests with a DS 1202 MicroLabBox show that the adaptive controller meets the desired performance requirements.

This paper is divided into five sections: The introduction is given in the first section. Section 2 aims to develop a mathematical model of the studied system. Section 3 is devoted

to developing the analysis of the adaptive backstepping technique. Section 4 is dedicated to the controller stability and tracking performances, which are validated by the simulations' results. The experimental results are presented in Section 5. Finally, the full text ends with the conclusion and references.

## 2. SP-PMSM Mathematic Model

Using the d-q representation of the SP-PMSM, the mathematical model of the SP-PMSM can be provided [20–25]:

$$\frac{di_d}{dt} = \frac{1}{L_d}\left[-R_s i_d + pL_q i_q \Omega_r + v_d\right], \tag{1}$$

$$\frac{di_q}{dt} = \frac{1}{L_q}\left[-R_s i_q - pL_d i_d \Omega_r - p\sqrt{\frac{3}{2}}\psi_{sf}\Omega_r + v_q\right], \tag{2}$$

$$\frac{d\Omega_r}{dt} = \frac{1}{J}\left[p\sqrt{\frac{3}{2}}\psi_{sf} i_q + p(L_d - L_q)i_d i_q - f\Omega_r - C_r\right], \tag{3}$$

Such that $v_d$, $v_q$ and $i_d$, $i_q$ are the d-axes and q-axes stator voltages and currents, respectively; $\Omega_r$ is the rotor's angular speed; $\psi_{sf}$ is the flux linkage; $p$ is the number of pole pairs of the SP-PMSM; $L_d$ and $L_q$ are the d-axes and q-axes stator inductances; the applied load-torque disturbance is represented by $C_r$; J is the rotor inertia; $R_s$ is the stator resistance; and $f$ is the viscous friction coefficient.

According to (1)–(3), the state-space model of the SP-PMSM may be expressed in the following manner:

$$\dot{x}_1 = \frac{1}{L_d}\left[-R_s x_1 + pL_q x_2 x_3 + u_1\right], \tag{4}$$

$$\dot{x}_2 = \frac{1}{L_q}\left[-R_s x_2 - pL_d x_1 x_3 - p\sqrt{\frac{3}{2}}\psi_{sf} x_3 + u_2\right], \tag{5}$$

$$\dot{x}_3 = \frac{1}{J}\left[p\sqrt{\frac{3}{2}}\psi_{sf} x_2 + p(L_d - L_q)x_1 x_2 - f x_3 - C_r\right], \tag{6}$$

where $x_1 = i_d$, $x_2 = i_q$, $x_3 = \Omega_r$, $u_1 = v_d$, and $u_2 = v_q$. The variables $u_1$ and $u_2$ are the actual control and $x$ is the state vector.

The main control objectives are:

- Controlling the signals $x_1$ and $x_3$ to their references;
- Estimating the non-measurable parameters of a SP-PMSM, such us $f$ and $J$;
- Estimating the load torque $C_r$.

Its objectives will be achieved to ensure a global asymptotic convergence of the system.

## 3. Adaptive Nonlinear Control

### 3.1. Adaptive Backstepping Design

The nonlinear adaptive controller is designed based on the backstepping technique, which is a recursive design methodology described in [21,26–28]. The main idea of backstepping is to let certain states act as "virtual commands". It involves a systematic construction of both feedback control laws and associated Lyapunov functions. The controller design is completed in a number of steps that is never higher than the system order.

In each step, the objective is to stabilize the system error to the origin using the Lyapunov theory.

This technique consists in finding a stabilizing function, which is a virtual control for each subsystem, based on the Lyapunov stability theory, until the overall control to the system can be determined.

In the last step, the control law is finally obtained.

### 3.2. Current-Controller Design Technique

The first goal is to eliminate the reluctance effect caused by $L_q \neq L_d$. Thus, the actual variable $x_1$ is forced to zero. For this purpose, a first error $z_1$ is introduced by:

$$z_1 = x_1 - x_1^* \; ; \; \text{with } x_1^* = 0, \tag{7}$$

such that the reference d-axis current is $x_1^*$.

The derivative of $z_1$ is given by:

$$\dot{z}_1 = \dot{x}_1 - 0 = \frac{1}{L_d}\big[-R_s x_1 + pL_q x_2 x_3 + u_1\big]. \tag{8}$$

The first Lyapunov function will now be discussed:

$$V_1 = \frac{1}{2}z_1^2 > 0. \tag{9}$$

Using (8), the time-derivative of the Lyapunov function is:

$$\dot{V}_1 = z_1 \dot{z}_1 = \frac{z_1}{L_d}\big(-R_s x_1 + pL_q x_2 x_3 + u_1\big). \tag{10}$$

This equation suggests the following choice of $\dot{z}_1$ :

$$\dot{z}_1 = -c_1 z_1 \; ; \; \text{with } c_1 > 0, \tag{11}$$

such that $c_1$ is a design parameter.

$$\dot{V}_1 = \dot{z}_1 z_1 = -c_1 z_1^2 < 0, \tag{12}$$

which shows that the error $z_1$ is exponentially disappearing ($z_1 \to 0$). Hence, it demonstrates that the equilibrium ($z_1 = 0$) is globally asymptotically stable (GAS).

The d-axis current-control law is then created by fusing (8) and (11):

$$u_1 = -L_d c_1 z_1 + R_s x_1 - pL_q x_2 x_3. \tag{13}$$

### 3.3. Speed-Controller Design Technique

The objective of the control is to make sure that the motor speed $x_3$ tracks the intended value in the presence of unknown parameters of the SP-PMSM and variation load torque. Again, the controller is built with a two-step adaptive backstepping method.

Step1:

First, the second tracking error is defined:

$$z_2 = x_3 - x_3^* \; ; \; \text{where } x_3^* \text{ is the speed reference.} \tag{14}$$

Deriving (14), it follows from (6) that:

$$\dot{z}_2 = \dot{x}_3 - \dot{x}_3^* = \frac{1}{J}\Big[p\Big(\psi + L_{dq}x_1\Big)x_2 - fx_3 - C_r\Big] - \dot{x}_3^*, \tag{15}$$

where $x_3^* = w_{ref}^*; L_{dq} = (L_d - L_q); \psi = \sqrt{\frac{3}{2}}\psi_{sf}$.

In Equation (15), the quantity $x_2$ stands as a virtual control variable. To determine the trajectory of this virtual control variable, the following second Lyapunov function is considered:

$$V_2 = \frac{1}{2}z_2^2 > 0. \tag{16}$$

The time-derivative of $V_2$ along the trajectory of (15) is given by:

$$\dot{V}_2 = z_2 \left[ \frac{1}{J} \left[ p \left( \psi + L_{dq} x_1 \right) x_2 - f x_3 - C_r \right] - \dot{x}_3^* \right].$$ (17)

Equation (17) shows that the tracking error $z_2$ can be regulated to zero if its dynamic is chosen as follows:

$$\dot{z}_2 = -c_2 z_2,$$ (18)

where $c_2 > 0$ is a second design parameter. Combining (15) and (18), the following stabilizing function is then defined by:

$$p \left( L_{dq} x_1 + \psi \right) x_2 = J \left( \dot{x}_3^* - c_2 z_2 \right) + f x_3 + C_r,$$ (19)

where $p \left( L_{dq} x_1 + \psi \right) x_2$ is the virtual control input variable. Since $p \left( L_{dq} x_1 + \psi \right) x_2$ is not the effective control input, the following stabilizing function is then defined by:

$$\alpha = J \left( \dot{x}_3^* - c_2 z_2 \right) + f x_3 + C_r.$$ (20)

This stabilizing function acts as the reference q-axis current.

On the other hand, parameters $J$ and $f$ are non-measurable, and the resistive torque $C_r$ depends on the load variations or the trajectory of the vehicle, which is unknown.

These parameters will be estimated by the adaptive backstepping control, which will increase the robustness and stability of the system:

$$\alpha = \hat{J} \left( \dot{x}_3^* - c_2 z_2 \right) + \hat{f} x_3 + \hat{C}_r,$$ (21)

where $\hat{J}, \hat{f}$ and $\hat{C}_r$ are the estimates of $J, f$ and $C_r$, respectively. We retain the expression of $\alpha$ and introduce a new command error. Hence, the following third error $z_3$ is defined by:

$$z_3 = \alpha - p \left( L_{dq} x_1 + \psi \right) x_2.$$ (22)

Note: in a steady state, $x_1$ tends to zero.

The following step is to establish the control-signal variation law $u_2$.

Using (21) and (22), Equation (15) becomes:

$$\dot{z}_2 = -c_2 z_2 - \frac{\tilde{J}}{J} \left( \dot{x}_3^* - c_2 z_2 \right) - \frac{\tilde{f}}{J} x_3 - \frac{\tilde{C}_r}{J} - \frac{z_3}{J},$$ (23)

where $\tilde{J}, \tilde{f}$ and $\tilde{C}_r$ are the errors of $J, f$ and $C_r$, respectively, and $\tilde{J} = J - \hat{J}, \tilde{f} = f - \hat{f}$ and $\tilde{C}_r = C_r - \hat{C}_r$.

Equation (17) of the Lyapunov function is rewritten as:

$$\dot{V}_2 = -c_2 z_2^2 - \left[ \frac{\tilde{J}}{J} \left( \dot{x}_3^* - c_2 z_2 \right) + \frac{\tilde{f}}{J} x_3 + \frac{\tilde{C}_r}{J} + \frac{z_3}{J} \right] z_2.$$ (24)

Step2:

From (23), the time-derivative of $z_3$, using (5), (11) and (21), is obtained as follows:

$$\dot{z}_3 = \hat{J} \left( \dot{x}_3^* - c_2 z_2 \right) + \hat{f} x_3 + \hat{C}_r + \hat{J} \ddot{x}_3^* + \hat{f} \dot{x}_3^* - \left( \hat{f} - c_2 \hat{J} \right) \left[ c_2 z_2 + \frac{\tilde{J}}{J} \left( \dot{x}_3^* - c_2 z_2 \right) + \frac{\tilde{f}}{J} x_3 + \frac{\tilde{C}_r}{J} + \frac{z_3}{J} \right] + c_1 p x_2 L_{dq} z_1 - \frac{p}{L_q} \left( L_{dq} x_1 + \psi \right) \left[ -R_s x_2 - p L_d x_1 x_3 - p \psi x_3 + u_2 \right].$$ (25)

We are finally able to make a convenient choice of the control signal $u_2$ to stabilize the whole system with the state vector $(z_2, z_3)$. To this end, we consider the augmented Lyapunov function candidate:

$$V_3 = V_2 + \frac{1}{2}z_3^2 + \frac{1}{2}\frac{1}{\gamma_1}\frac{\widetilde{J}^2}{J} + \frac{1}{2}\frac{1}{\gamma_2}\frac{\widetilde{C}_r^2}{J} + \frac{1}{2}\frac{1}{\gamma_3}\frac{\widetilde{f}^2}{J} > 0,$$  (26)

where $\gamma_1$, $\gamma_2$ and $\gamma_3$ are, respectively, the design parameters. Its time-derivative, using Equation (25), is given by:

$$\dot{V}_3 = \dot{V}_2 + \dot{z}_3 z_3 + \frac{\dot{\widetilde{J}}}{\gamma_1}\frac{\widetilde{J}}{J} + \frac{\dot{\widetilde{C}}_r}{\gamma_2}\frac{\widetilde{C}_r}{J} + \frac{\dot{\widetilde{f}}}{\gamma_3}\frac{\widetilde{f}}{J}.$$  (27)

Using (24) and (25), Equation (26) becomes:

$$\dot{V}_3 = -c_2 z_2^2 - c_3 z_3^2 + \frac{\widetilde{J}}{J}\left[\frac{\dot{\widetilde{J}}}{\gamma_1} + \left(c_2 z_2 - \dot{x}_3^*\right)\left(z_2 + \left(\hat{f} - c_2\hat{J}\right)z_3\right)\right] + \frac{\widetilde{C}_r}{J}\left[\frac{\dot{\widetilde{C}}_r}{\gamma_2} - \left(z_2 + \left(\hat{f} - c_2\hat{J}\right)z_3\right)\right] +$$
$$\frac{\widetilde{f}}{J}\left[\frac{\dot{\widetilde{f}}}{\gamma_3} - \left(z_2 + \left(\hat{f} - c_2\hat{J}\right)z_3\right)x_3\right].$$  (28)

We assume

$$\left[\begin{array}{l}\hat{J}\left(\dot{x}_3^* - c_2 z_2\right) + \hat{f}x_3 + \hat{C}_r + \hat{J}\ddot{x}_3^* + \hat{f}\dot{x}_3^* - \frac{z_2}{\hat{J}} - \left(\hat{f} - c_2\hat{J}\right)\left(\frac{z_3}{\hat{J}} + c_2 z_2\right) \\ +c_1 p x_2 L_{dq} z_1 - \frac{p}{L_q}\left(L_{dq}x_1 + \psi\right)\left(-R_s x_2 - pL_d x_1 x_3 - p\psi x_3 + u_2\right)\end{array}\right] = -c_3 z_3,$$  (29)

where $c_3 > 0$ is a third design parameter.

However, Equation (28) suggests that the control signal $u_2$ should be chosen in a way that the three terms within the braces are set to zero.

This helps us to propose the adaptive control theory for the purpose of estimating the three parameters by:

$$\dot{\widetilde{J}} = -\lambda_J,$$  (30)

$$\dot{\widetilde{C}}_r = -\lambda_{C_r},$$  (31)

$$\dot{\widetilde{f}} = -\lambda_f,$$  (32)

or

$$\lambda_J = \left(c_2 z_2 - \dot{x}_3^*\right)\left(z_2 + \left(\hat{f} - c_2\hat{J}\right)z_3\right)\gamma_1,$$  (33)

$$\lambda_{C_r} = -\left(z_2 + \left(\hat{f} - c_2\hat{J}\right)z_3\right)x_3\gamma_2,$$  (34)

$$\lambda_f = -\left(z_2 + \left(\hat{f} - c_2\hat{J}\right)z_3\right)\gamma_3.$$  (35)

From (23) and (30)–(35), the estimate parameters of the SP-PMSM $\hat{J}$, $\hat{f}$, and the estimate of load torque $\hat{C}_r$ are calculated using Equations (36)–(38):

$$\dot{\hat{J}} = \lambda_J,$$  (36)

$$\dot{\hat{C}}_r = \lambda_{C_r},$$  (37)

$$\dot{\hat{f}} = \lambda_f.$$  (38)

Therefore, using (30)–(35), Equation (28) becomes

$$\dot{V}_3 = -c_2 z_2^2 - c_3 z_3^2 < 0. \tag{39}$$

Equation (39) shows that the equilibrium $(z_2, z_3) = (0, 0)$ is globally asymptotically stable.

Hence, from (29), the following second control law is given by

$$u_2 \frac{L_q}{p\left(L_{dq}x_1 + \psi\right)}\left[ \begin{array}{c} \dot{\hat{J}}\left(\dot{x}_3^* - c_2 z_2\right) + \dot{\hat{f}}x_3 + \dot{\hat{C}}_r + \hat{J}\ddot{x}_3^* + \hat{f}\dot{x}_3^* - \left(\hat{f} - c_2\hat{J}\right)\left(\frac{z_3}{\hat{J}} + c_2 z_2\right) + c_1 p x_2 L_{dq} z_1 \\ -\frac{z_2}{\hat{J}} + \frac{p}{L_q}\left(L_{dq}x_1 + \psi\right)R_s x_2 + c_3 z_3 + \frac{p}{L_q}\left(L_{dq}x_1 + \psi\right)p(L_d x_1 + \psi)x_3 \end{array} \right]. \tag{40}$$

The following statement encapsulates the paper's key finding:

**Proposition 1.** *If the closed-loop system consisting of a SP-PMSM system is represented by (4)–(6), and the controller is composed of the control laws (13) and (40), then the following must take place:*

i.　　Closed-loop system is GAS;
ii.　　Current d-axis regulation is set to zero;
iii.　　Perfect tracking of motor speed $x_3$ is set to its reference;
iv.　　Non-measurable parameters of the SP-PMSM such as $f$ and $J$ are estimated;
v.　　Load torque $C_r$ is estimated.

**Proof.** The following global Lyapunov function should be defined by $V$:

$$V = V_1 + V_3 = \frac{1}{2}z_1^2 + \frac{1}{2}z_2^2 + \frac{1}{2}z_3^2 + \frac{1}{2}\frac{1}{\gamma_1}\frac{\tilde{J}^2}{J} + \frac{1}{2}\frac{1}{\gamma_2}\frac{\tilde{C}^2_r}{J} + \frac{1}{2}\frac{1}{\gamma_3}\frac{\tilde{f}^2}{J}. \tag{41}$$

Its time-derivative, using (12) and (40), is obtained as follows:

$$\dot{V} = -c_1 z_1^2 - c_2 z_2^2 - c_3 z_3^2. \tag{42}$$

From (41) and (42), one has $V$ positive-definite and $\dot{V}$ negative-definite, which clearly means that the equilibrium $(z_1, z_2, z_3) = (0, 0, 0)$ is globally asymptotically stable (see, e.g., [29]). As a result, all tracking faults gradually disappear. The evidence of the proposition is now complete. □

The suggested controller will be simulated using MATLAB®/Simulink® in the next section.

## 4. Numerical Simulation

This simulation is used to demonstrate how well the suggested adaptive backstepping controller performs. Table 1 lists the system characteristics; these values are taken from paper [17]. Using the MATLAB®/Simulink® software, the SP-PMSM control-simulation bench is created, as shown in Figure 1.

The following design-control parameters are chosen: $c_1 = 20$, $c_2 = 2 \times 10^3$, $c_3 = 200$, $\gamma_1 = 3 \times 10^{-3}$, $\gamma_2 = 5 \times 10^{-3}$ and $\gamma_3 = 7 \times 10^{-3}$. Theoretically, the design parameters must only be positive. The achieved transient performances are determined by these values.

The point is that, as it is generally the case in nonlinear control design, there is no systematic rule for conveniently selecting these numerical values. The usual practice is to use the 'trial-and-error' method, which consists in progressively increasing the parameter values until a satisfactory compromise is achieved between the rapidity of responses and the control activity.

**Table 1.** Parameters of the controlled system.

| Parameter | Value |
|---|---|
| Stator resistance $R_s$ | 0.56 Ω |
| Number of pole pairs $p$ | 3 |
| Rotation inertia $J$ | 0.0021 kg.m$^2$ |
| Flux of permanent magnet $\psi_{sf}$ | 0.82 Wb |
| Inductance $L_d$ | 0.048 H |
| Inductance $L_q$ | 0.064 H |
| Viscous damping $f$ | 0.0001 Nm/rd.s$^{-1}$ |
| Rated voltage | 320 V |
| Rated power | 2 kW |
| Rated speed | 1800 r/mn |

The simulation in this paper will be divided into 4 parts to show the robustness of the proposed control system. In this paper, the direct-axis current $i_d$ is always set to zero to remove the effect of the torque $i_d \times i_q$ in Equation (3). Hence, the robustness of our controller is ensured.

### 4.1. Speed Change

The objective of this simulation is to check the tracking behavior of the proposed controller under the European EUDC (Extra Urban Driving Cycle). The latter allows us to constitute a real test to assess the effectiveness of the proposed controllers in automotive applications [30].

Firstly, we shall consider that the internal variables of the SP-PMSM ($J$ and $f$) and the load torque $C_r$ are constants ($C_r = 5 \, Nm$).

Figure 2 illustrates the vehicle speed and its reference. The speed trajectory is perfectly tracked thanks to the proposed controller, as stated by Proposition 1.

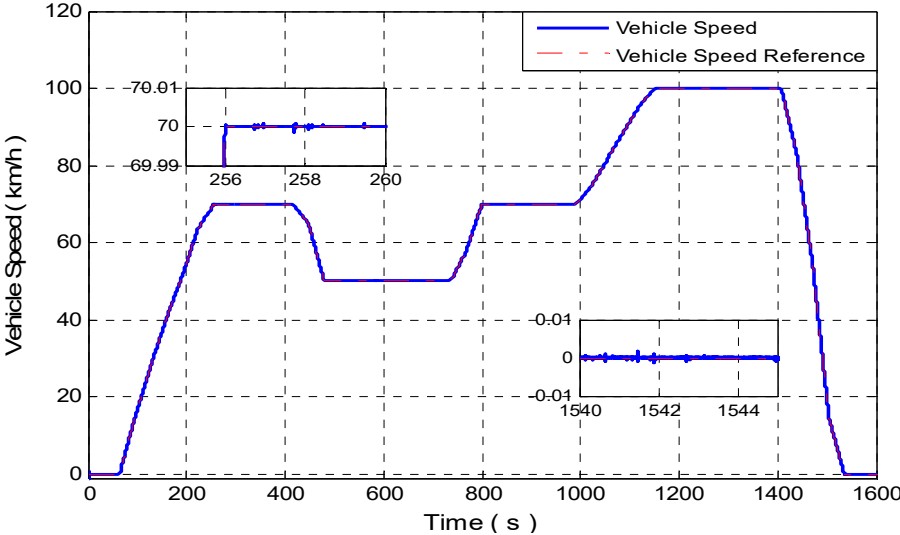

**Figure 2.** Vehicle speed and its reference, with zoom.

Figure 3 illustrates the load torque and the electromagnetic torque. The zoom presents that the gap between the load torque and the electromagnetic torque denotes the loss-torque $C_p = f.\Omega$, according to this equation: $C_{em} = J\frac{d\Omega_r}{dt} + C_r + C_p$.

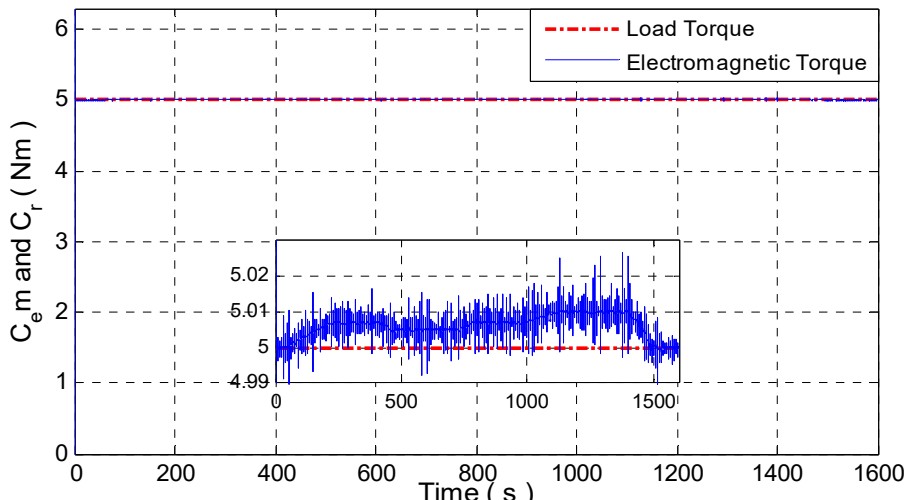

**Figure 3.** Load torque and electromagnetic torque, with zoom.

Figure 4 illustrates the current $i_q$ and $i_d$. The current $i_q$ comes to confirm our interpretation, because $p.\sqrt{\frac{3}{2}}\psi_{sf}.i_q = f.\Omega + C_r$, where $p$, $\psi_{sf}$ and $C_r$ are constants. Hence, a large variation in $\Omega$ generates a small variation in $i_q$. Therefore, this gap represents the mechanical losses of the SP-PMSM. The current $i_d$ is always set to zero to remove the effect of the torque $i_d \times i_q$ in Equation (3).

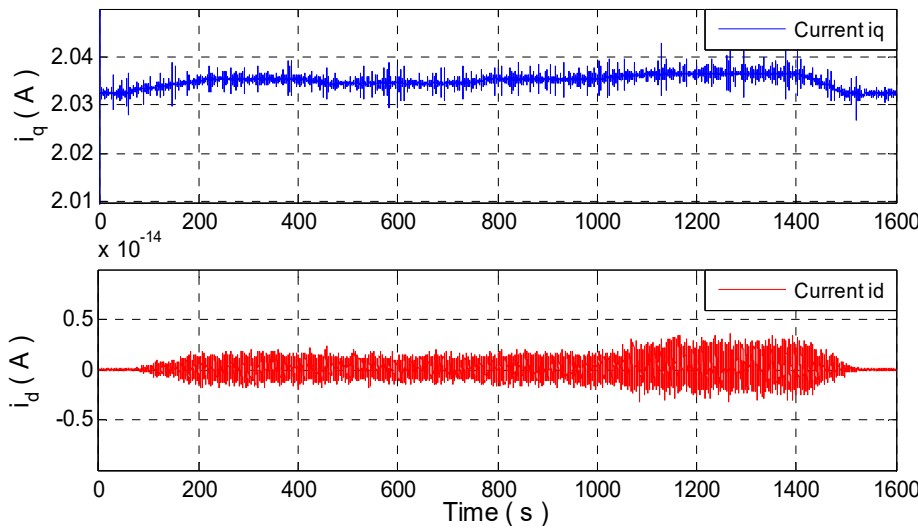

**Figure 4.** Current $i_q$ and current $i_d$.

From this experiment, we can recapitulate:

- The proposed control guarantees perfect vehicle speed tracking to its reference;
- The closed-loop system is stable with respect to the variation in the reference speed.

### 4.2. Inertia Change

In this part, we will see the effect of inertia on vehicle speed, while keeping the load torque $C_r$, the friction $f$ and the reference velocity constant.

Figure 5 illustrates the estimated inertia and its reference value and shows a zoomed view of both signals. We varied the value of inertia at 5 s and then at 10 s by 143% and 190%, respectively, compared to its initial value. One can see from this figure that the estimated inertia perfectly tracks its reference without overshoot, with a response time not exceeding 50 ms, which shows the performance of the proposed controller.

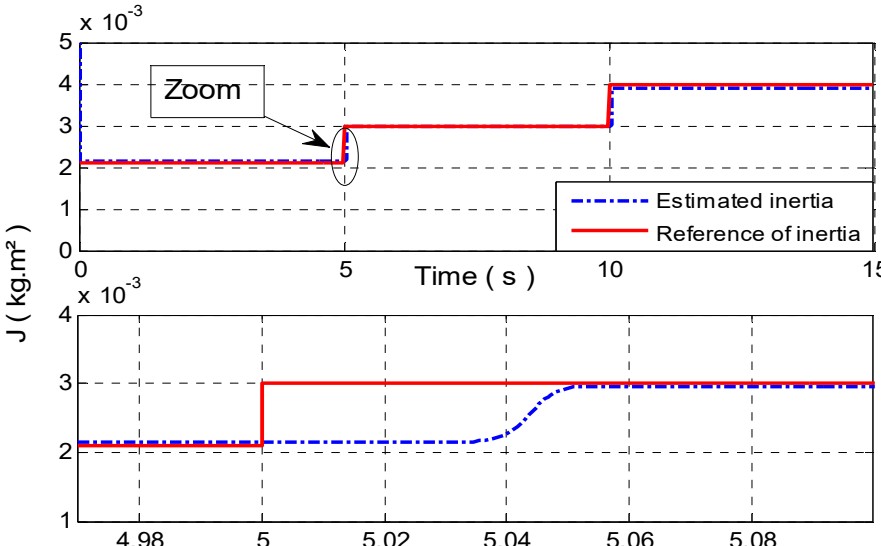

**Figure 5.** The estimated inertia and its reference value with zoom.

Figure 6 clearly shows that the vehicle speed remains constant while changes in inertia take place. The instants of variation of the inertia generate a small variation in the vehicle speed for 20 ms.

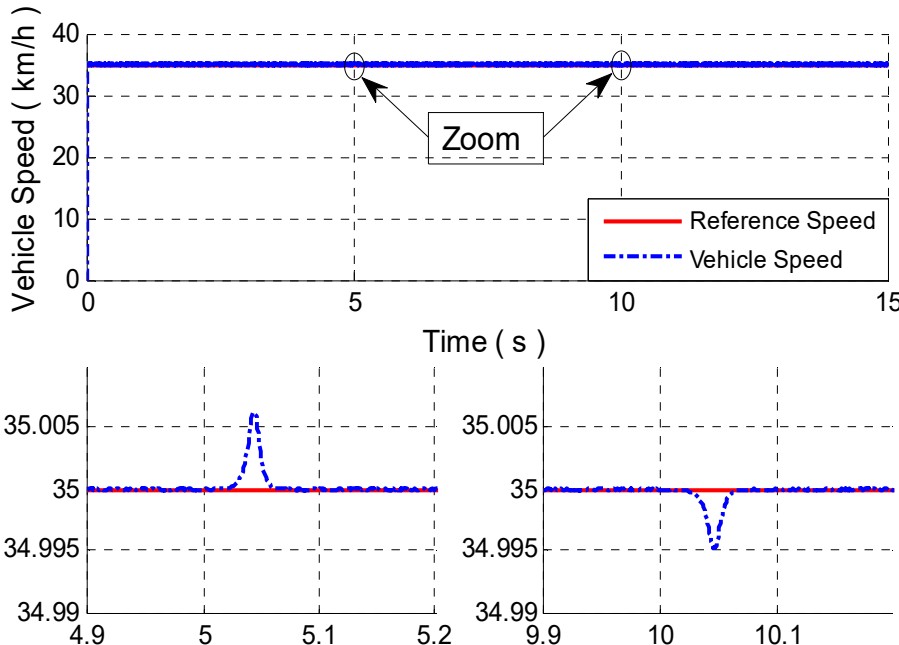

**Figure 6.** Vehicle speed and its reference with zoom.

Figure 7 illustrates that the electromagnetic torque remains constant because $C_{em} = J\frac{d\Omega_r}{dt} + C_r + C_p$, or the speed of rotation $\Omega_r$ is controlled at its reference, which derivative is zero. Therefore, the electromagnetic torque $C_{em}$ is constant. The zoom presents that the gap between the load torque and the electromagnetic torque denotes the loss-torque $C_p = f.\Omega$.

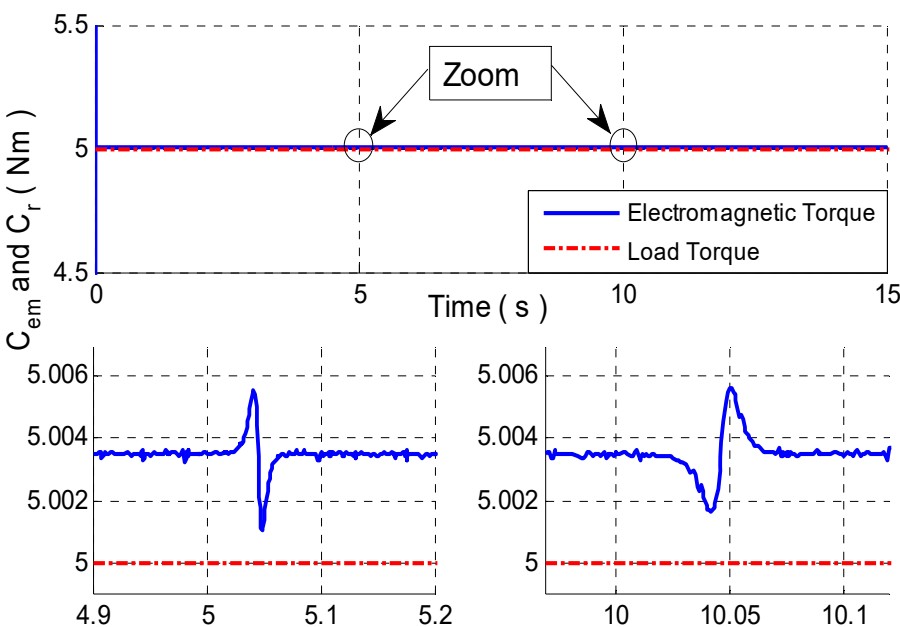

**Figure 7.** Electromagnetic torque with zoom.

Figure 8 shows that the current $i_q$ remains constant. It depends on the load torque, which means that the variation in SP-PMSM power is almost zero.

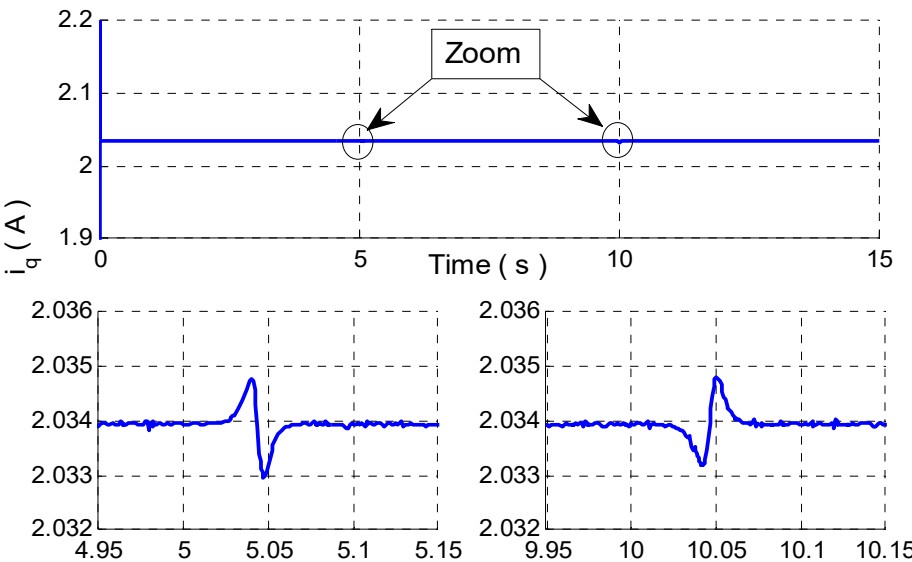

**Figure 8.** Current i$_q$, with zoom.

Figure 9 shows that the current $i_d$ is always set to zero to remove the effect of the torque $i_d \times i_q$ in Equation (3).

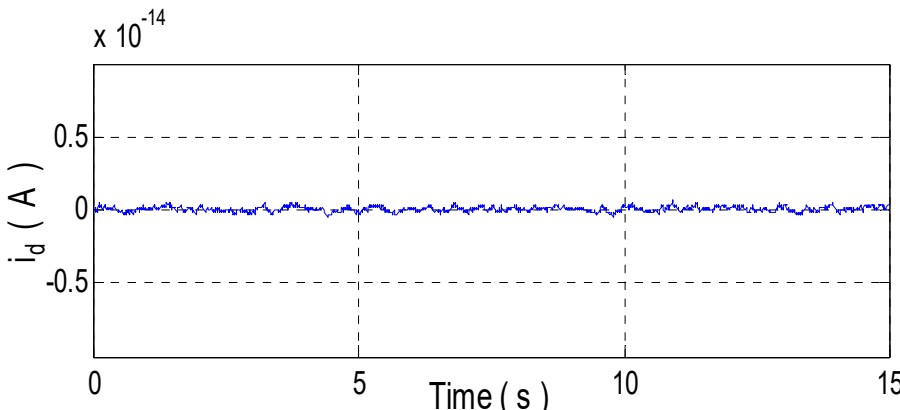

**Figure 9.** Current $i_d$.

From this experiment, we can recapitulate:

- The perfect tracking of the vehicle speed to its reference;
- The estimation of inertia $J$ by this controller helps to ensure the stability and robustness of the system;
- The moment of inertia $J$ depends on the mass but not on the speed.

### 4.3. Friction Change

In the following cases, we will fix the inertia $J$, load torque $C_r$ and the reference speed. Thereafter, we will see the effect of friction on the vehicle speed.

Figure 10 illustrates the estimated friction and its reference value. We can see that the estimated friction perfectly tracks its reference. It shows a good performance of the controller, as it is shown in the zoomed view of the two signals.

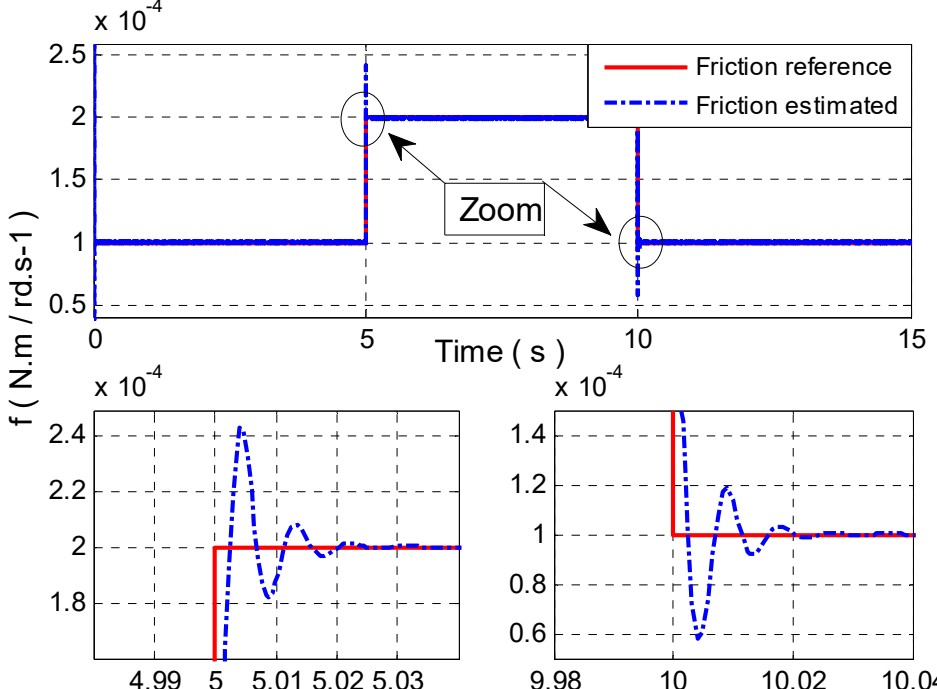

**Figure 10.** Estimated friction and its reference value, with zoom.

The overshooting occurring at the transient moment represents 20% of the reference value, which is an acceptable value where the response time is 20 ms.

Figure 11 clearly shows that the vehicle speed remains constant while changes in friction take place. This variation in friction generates a small variation in the vehicle speed for 20 ms.

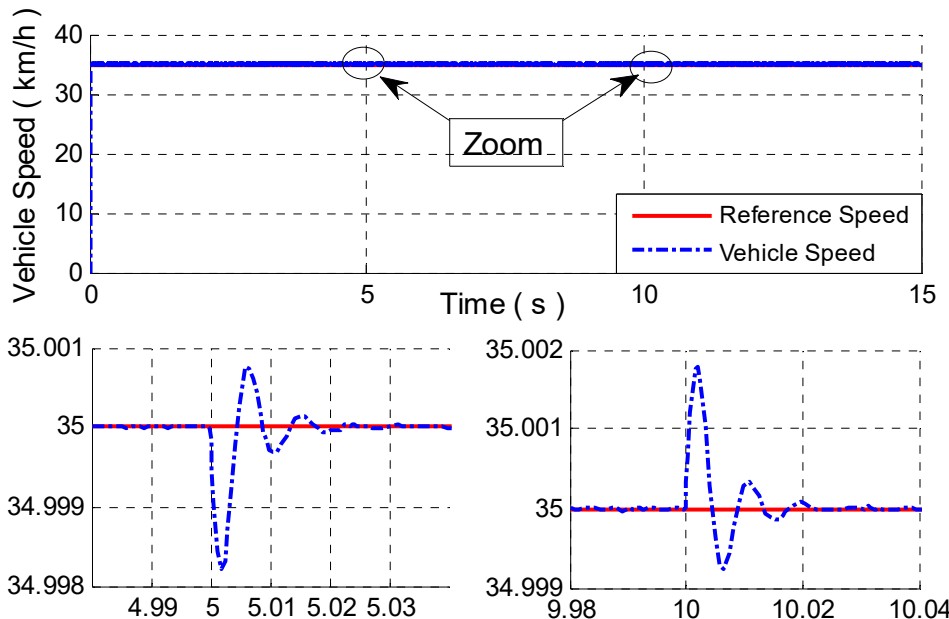

**Figure 11.** Vehicle speed and its reference with zoom.

Figure 12 illustrates the load torque and the electromagnetic torque. The zoom view presents that the gap between the load torque and the electromagnetic torque denotes the loss-torque $C_p = f.\Omega$.

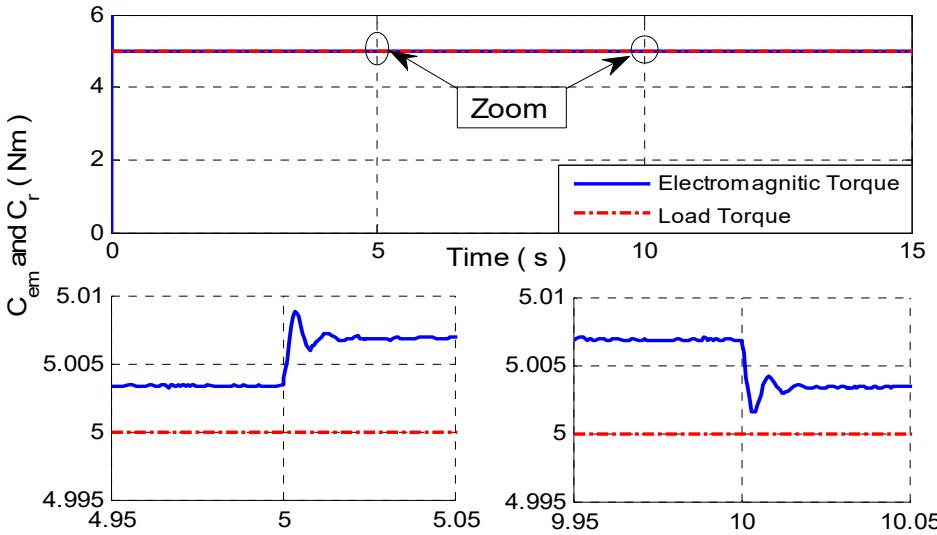

**Figure 12.** Load torque and electromagnetic torque with zoom.

Figure 13 represents the variation in the quadratic current $i_q$, which leads to an increase in the power drops of the SP-PMSM.

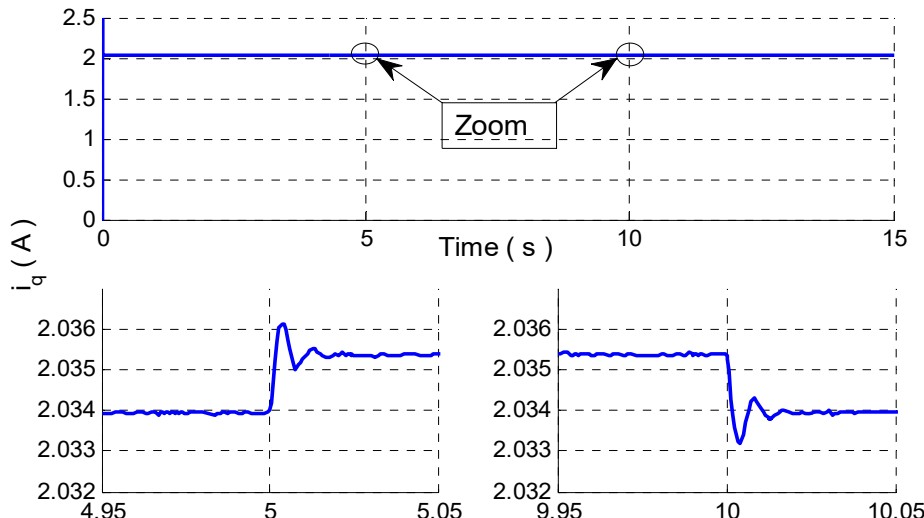

**Figure 13.** Current i$_q$ with zoom.

Figure 14 illustrates that the current $i_d$ is always set to zero to remove the effect of the torque $i_d \times i_q$ in Equation (3).

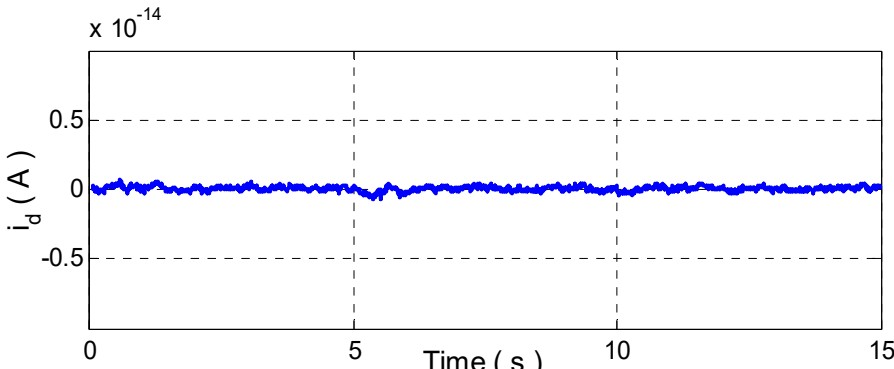

**Figure 14.** Current i$_d$.

From this experiment, we highlight:

- The perfect tracking of the vehicle speed to its reference;
- The estimation of friction $f$ by this controller helps to ensure the stability and robustness of the system;
- The variation in friction due to the aging of the machine or in cases where the maintenance schedule of the machine is not maintained.

### 4.4. Torque Change

Finally, we will fix the internal parameters of the HEV and the reference speed. Thereafter, we will see the effect of the load torque $C_r$ on the vehicle speed.

Figure 15 illustrates the estimated load torque, its reference value and the electromagnetic torque. We can see that the estimated load torque perfectly tracks its reference, showing the good performance of the controller. The response time is linear with the variations in the load torque.

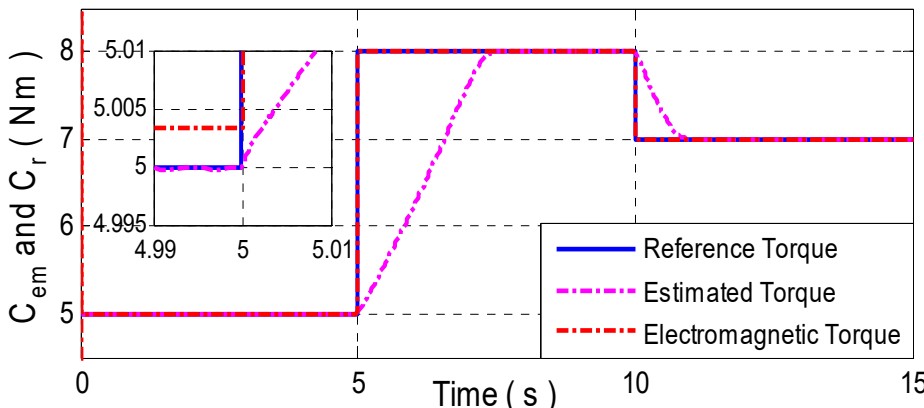

**Figure 15.** Estimated load torque, its reference value and electromagnetic torque.

Figure 16 illustrates the vehicle speed and its reference. We notice that the speed trajectory is perfectly tracked, which justifies the robustness of this control law. The overshooting occurring at the transient moment is almost linear with the load-torque variations.

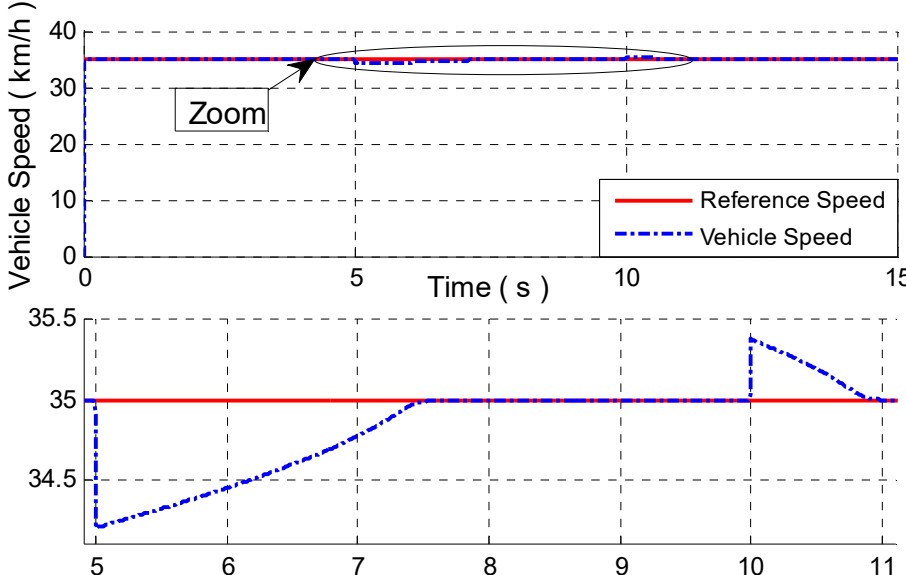

**Figure 16.** Vehicle speed and its reference with zoom.

Figure 17 shows that the magnitude of the q-axis current $i_q$ is dependent on the load torque (it varies according to the variations in load torque).

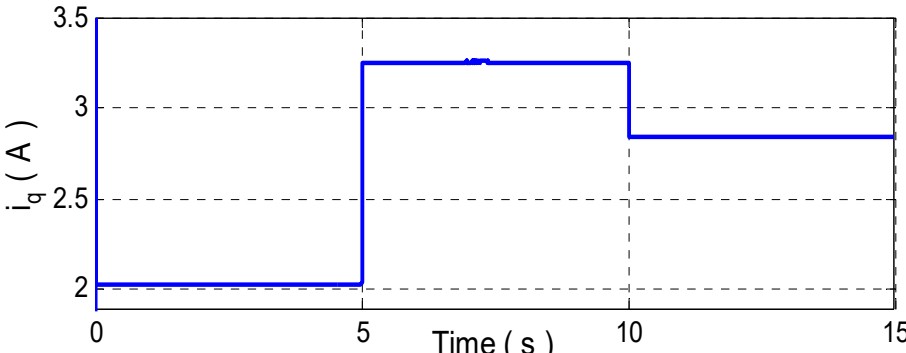

**Figure 17.** Current $i_q$.

Figure 18 illustrates that the current $i_d$ is always set to zero to remove the effect of the torque $i_d \times i_q$ in Equation (3).

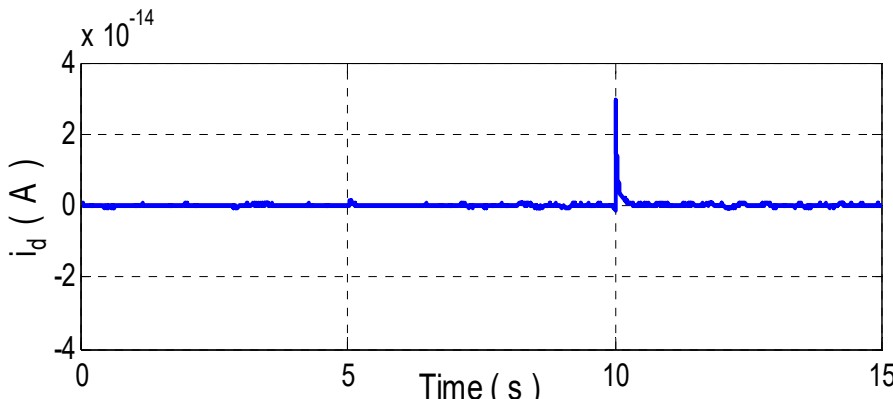

**Figure 18.** Current $i_d$.

From this experiment, we can recapitulate:

- The perfect tracking of the vehicle speed to its reference;
- The estimation of $C_r$ by this controller helps to ensure the stability and robustness of the system.

The next section will be devoted to the experimental tests of the proposed controller using the DS 1202 MicroLabBox.

## 5. Experimental Results

The experimental test bench of the salient-pole permanent-magnet synchronous motor system is described in Figure 19. Fuel cell was employed to have a substantial amount of power coming from the input source, and the control system was implemented using the dSPACE 1202 and Control Desk® software. The test bench consists essentially of:

- Three metal hydride canisters from Heliocentris with storage capacities of 800 NL of hydrogen;
- A Ballard Nexa 1200 fuel-cell module with its monitoring software;
- A power supply from BK Precision.
- A MicroLabBox-dSPACE DS1202 with Control Desk® software plugged into a Pentium 4 personal computer.
- A salient-pole permanent-magnet synchronous motor;
- A DC power supply;
- A DC/AC converter;
- A DC/DC converter;
- A Hall-Effect current sensor;
- A voltage sensor;
- A digital scope;
- A magnetic powder brake;
- An encoder position sensor;
- A torque sensor.

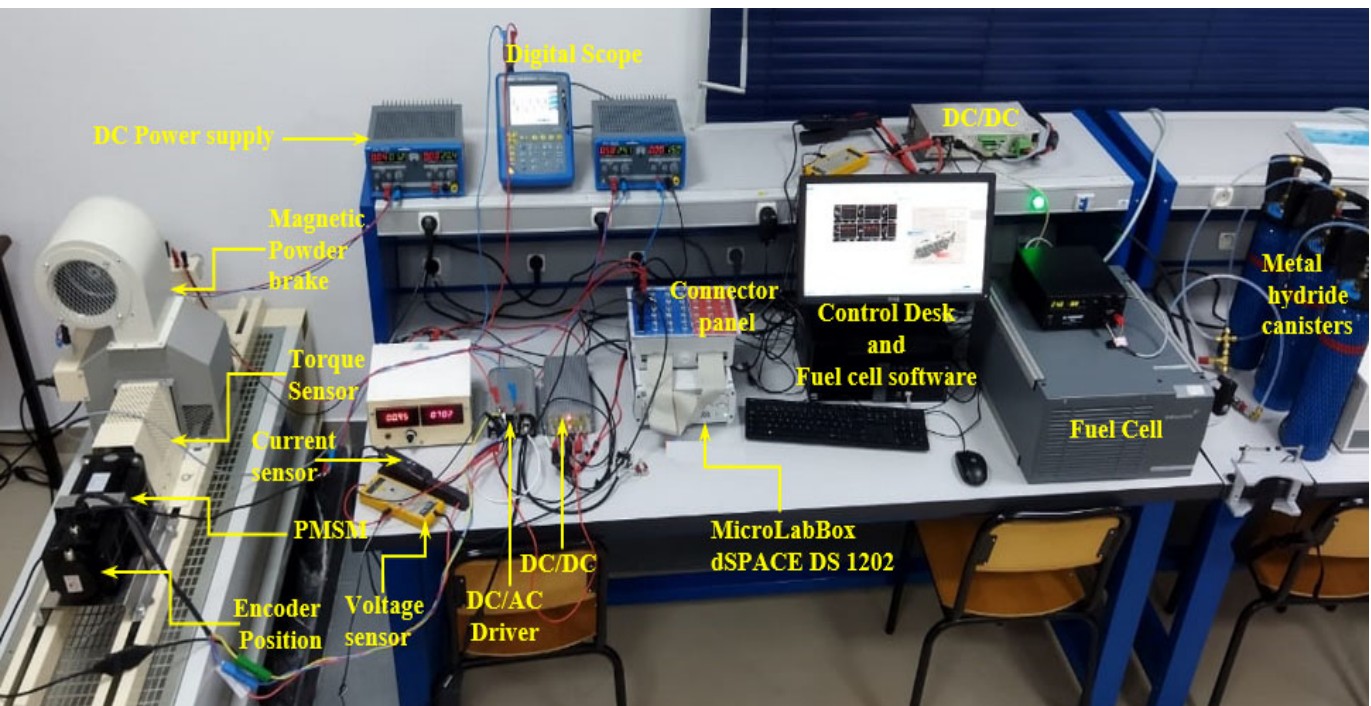

**Figure 19.** Laboratory prototype used for experimental validation.

Using Control Desk, the variations in the speed are programmed. The variations in the load torque are programmed by the DC power supply. The controlled-system characteristics are listed in Tables 2–6. The adaptive control-design parameters are shown in Table 7.

**Table 2.** Parameters of fuel DC/DC converter.

| Parameter | Value |
| --- | --- |
| Output power | 1200 W |
| Output current | max. 55 A |
| Nominal voltage | 24 V |
| Output voltage | 0–32 V |
| Input voltage | 16–45 V |
| Operational temperature | −10–55 °C |
| Efficiency | >96% |

**Table 3.** Parameters of the fuel-cell power module.

| Parameter | Value |
| --- | --- |
| Rated power | 1200 W |
| Rated current | 52 A |
| Rated voltage | 24 V |
| Output voltage | 20–36 V |
| Operational temperature | 5–40 °C |

**Table 4.** Parameters of the DC/DC converter.

| Parameter | Value |
| --- | --- |
| Output power | 2016 W |
| Output current | max. 42 A |
| Output voltage | 48 V |
| Input voltage | 24 V |

**Table 5.** Parameters of the PMSM.

| Parameter | Value |
|---|---|
| Rated power | 1000 W |
| Rated current | 25.7 A |
| Rated voltage | 48 V |
| Rated speed | 1000 r/min |
| Rated torque | 10 N.m |

**Table 6.** Parameters of PMSM Driver.

| Parameter | Value |
|---|---|
| Rated power | 2000 W |
| Rated voltage | 48 V |
| Continuous current | 60 A |
| Peak current | 150 A |
| Working frequency | 16.6 kHz |

**Table 7.** The design control parameters.

| Parameter | Value |
|---|---|
| $c_1$ | 20 |
| $c_2$ | 2000 |
| $c_3$ | 200 |
| $\gamma_1$ | 0.003 |
| $\gamma_2$ | 0.005 |
| $\gamma_3$ | 0.007 |

The experimental test-bench results will be divided into two parts to show the objectives of the proposed control system.

### 5.1. Speed Change

The speed reference signal switches from 600 r/min to 800 r/min at 30.79 s, and returns to 700 r/min at 60.79 s. The load torque is set to $C_r = 1$ Nm.

### 5.2. Torque Change

The speed reference signal is set to 700 r/min. The load torque switches from 5 Nm to 0.3 Nm at 30.89 s, and returns to 2.57 Nm at 61.29 s.

The experimental results for the adaptive backstepping controller of the SP-PMSM system are shown in Figures 20–27. The figures clearly illustrate that the experimental results confirm the performances shown by the theoretical analysis and simulations. More precisely, the speed regulation and the stability of the system are ensured.

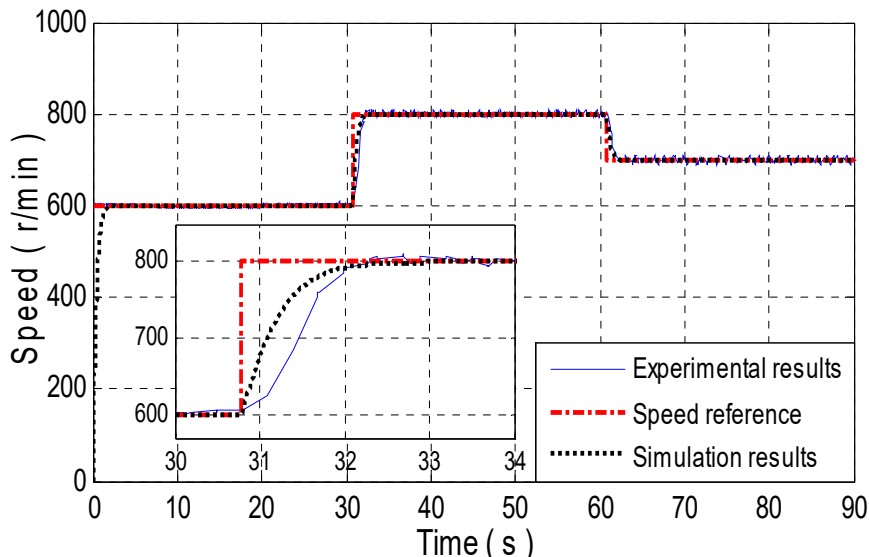

**Figure 20.** Speed profile and its reference, with zoom.

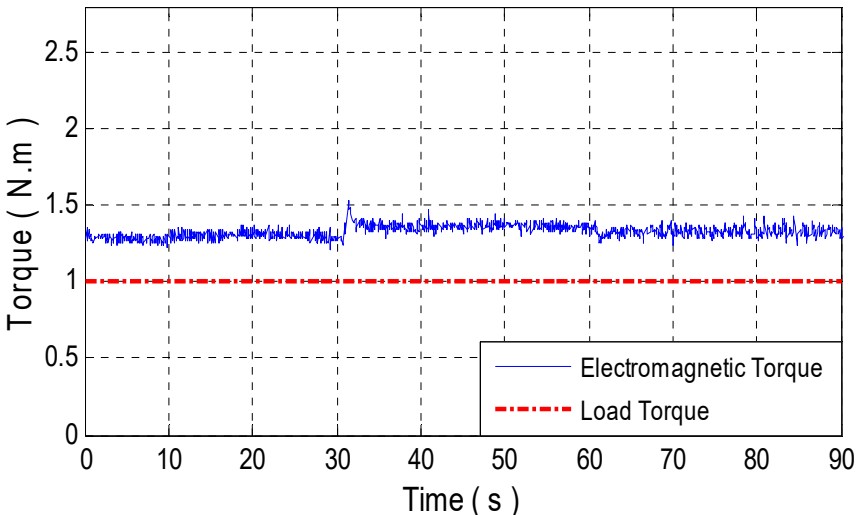

**Figure 21.** Load torque and electromagnetic torque.

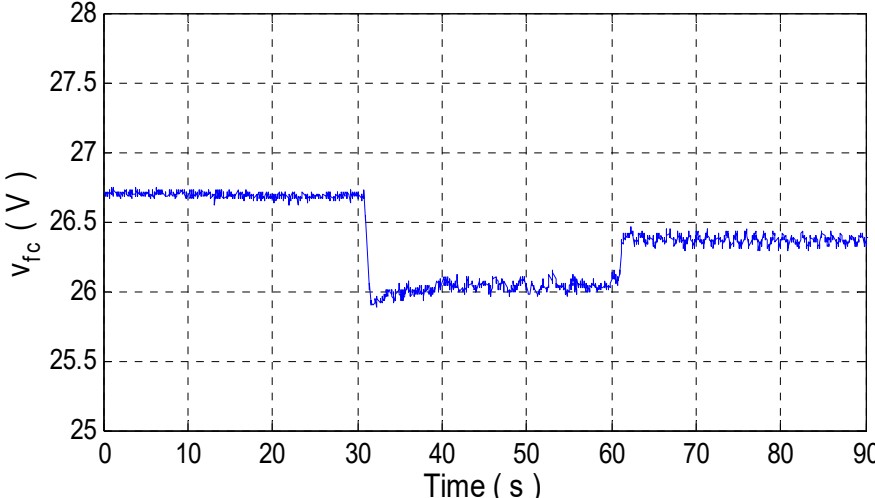

**Figure 22.** Fuel-cell voltage $V_{fc}$.

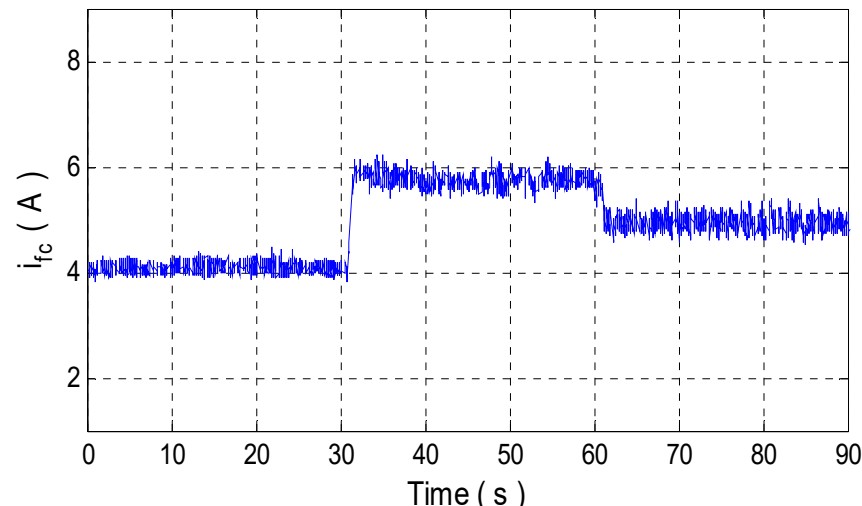

**Figure 23.** Fuel-cell current i$_{fc}$.

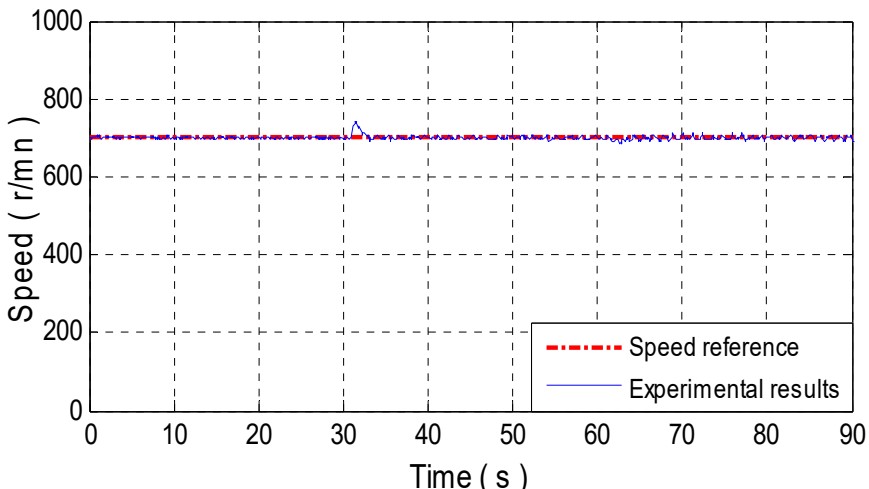

**Figure 24.** Speed and its reference.

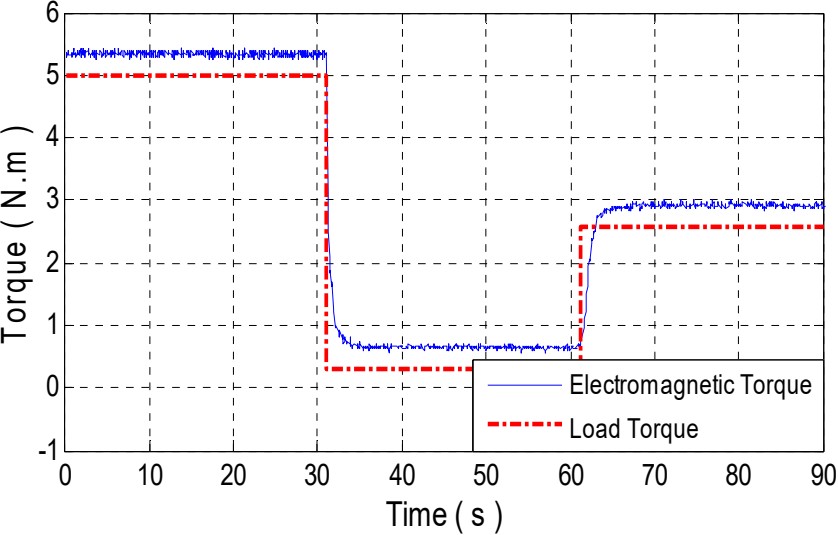

**Figure 25.** Load-torque profile and electromagnetic torque.

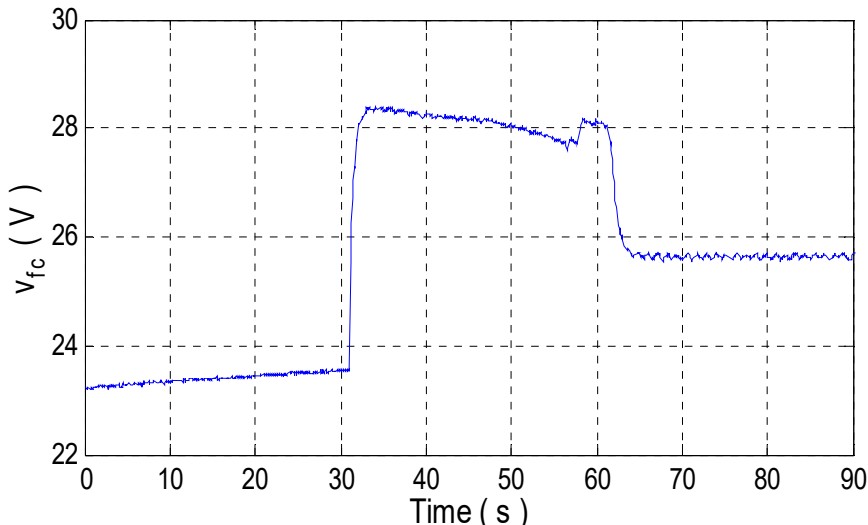

**Figure 26.** Fuel-cell voltage $V_{fc}$.

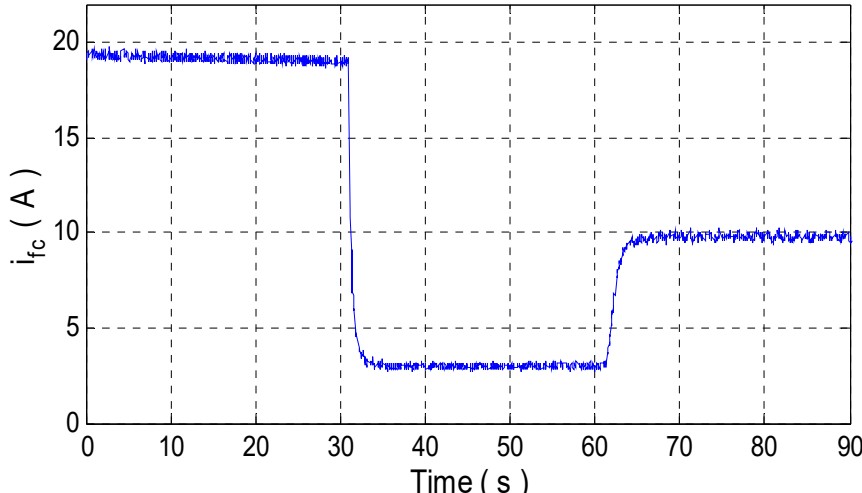

**Figure 27.** Fuel-cell current $i_{fc}$.

### 6. Conclusions

In this paper, a nonlinear adaptive regulator based on the backstepping technique is elaborated to control the speed of a salient-pole permanent-magnet synchronous motor to its reference, and to estimate the load torque $C_r$ and internal parameters (inertia and friction) of the SP-PMSM. Finally, according to the formal analysis, simulation and experimental results, it is shown that the obtained nonlinear adaptive controller ensures objectives such as:

- The perfect tracking of the vehicle speed to its reference;
- The high stability of the closed-loop system;
- The estimation of non-measurable parameters of the SP-PMSM such as f and J;
- The estimation of the load torque Cr.

These promising results can be drawn in future studies by comparing this control method with other existing controls, namely the sliding-mode control and PI.

**Author Contributions:** Methodology, Z.E.I. and H.E.F.; Software, C.E.F. and Z.E.I.; Validation, Z.E.I. and A.L.; Formal analysis, Z.E.I. and H.E.F.; Resources, C.E.F.; Writing—original draft, C.E.F.; Writing— review & editing, C.E.F., Z.E.I., A.L., F.Z.B., K.G. and A.R.; Visualization, F.Z.B., K.G. and A.R.; Supervision, H.E.F.; Funding acquisition, C.E.F. and Z.E.I. All authors have read and agreed to the published version of the manuscript.

**Funding:** This research received no external funding.

**Data Availability Statement:** Data available in a publicly accessible repository.

**Conflicts of Interest:** The authors declare no conflict of interest.

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
