# Peer review of "Adaptive Nonlinear Control of Salient-Pole PMSM for Hybrid Electric Vehicle Applications: Theory and Experiments"

_wevj, doi:10.3390/wevj14020030_

Round 1
Reviewer 1 Report
1- Check keywords: Salient-Pole Permanent synchronous motor magnetic (PMSM)
2- Re-arrange the literature survey.
3- Check for different fonts in the paper.
4- Check the location of Table 1.
5- Most of the given information in Table 1 is incorrect. Such as bldc cost vs pmsm cost, bldc controller cost vs pmsm controller cost, and total efficiency of bldc.
6- There is not enough info about pmsm drive used in the experimental study. Also silent pole pmsm motor geometry should be given.
7- The overall arrangement of the manuscript is poor. The main contributions of the paper are not given. Some results are given as what they should be but there is no connection between methods and it is hard to understand what has been achieved.
Author Response
The authors are grateful to the Reviewers for their helpful comments. The paper has been revised in accordance with Reviewer’s comments and suggestions. The changes made are described in the following responses to each Reviewer comment. The changes are highlighted in the paper using red colour.

Reviewer 2 Report
Very good work. It is a great contribution to the development of the transformation of the automotive industry towards hybrid and electric vehicles.
Author Response

(The authors gave the same response as above.)

Reviewer 3 Report
This is an interesting manuscript with a topic of controlling a Salient-Pole Permanent Magnet Synchronous Motor (PMSM) used in hybrid electric vehicles. An adaptive nonlinear controller based on Backstepping technique is developed. The paper has the following issues that need to be improved:
1. Line 14 on page 5 and line 9 on page 9 indicate that tracking error will disappear exponentially, which needs further explanation, and it can be better for supplement the curve of error changing.
2. What are the ‘design parameters’ related to in equation (22) on page 7and how to obtain the value of ‘design parameters’?
3. The simulation and experiment should provide the comparison with the previous control algorithm, mentioned in the abstract, to illustrate the superiority of the proposed algorithm?
4. The part of the experimental results analysis is not enough, and the part of the experimental results needs quantitative analysis of control algorithm (number, table and textual description).
5. how to measure and evaluate the designed controller to meet the robustness and reliability and stability requirement.
6. The conclusion section should be organized item by item, and demonstrate the robustness and reliability of the designed controller by number. Also the future research should be given.
7. Formulas and letters in the text need to be further checked to increase readability.
Author Response

(The authors gave the same response as above.)

Reviewer 4 Report
This paper proposes an adaptive control law for the speed control of a PMSM motor with uncertain parameters. It follows a conventional Lyapunov based adaptive control design, and it demonstrates the feasibility of the resulting controller in theory, in simulation, and in an experiment.
The paper is generally well written and easy to follow. The adaptive control, especially equation (29), does seem genuinely novel and well motivated. However, while the adaptive control design is done proficiently, the automotive application of the test is much less convincing. This makes it hard to come up with an overall verdict. From an adaptive control perspective, this is a sound paper, but from an automotive perspective, not so much.
The following improvements can be made:
The equations are not properly set - they are off the line, and the font is too small compared to the text font. Spacing is also off.
The table on page 3 has no caption, and it is not quite correct. Brushless DC and PMSM are just two different views on the same topology, so they cannot have different properties (but there are different versions of PMSM). The table is also missing Synchronous Reluctance, which is a very important emerging topology.
The assumption in (3) is not viable - i_d is a key variable for flux weakening control, which is essential for the efficient operation of the PMSM over the typical operating range of a vehicle. Setting i_d to zero makes this approach non-viable for automotive applications. But it seems appropriate in the context of the control design as a simplifying assumption.
(9) is interesting, because the feedforward term is commonly used in motor control (a phase advance to compensate for L_q), but the feedback term is potentially novel.
The problem with the numerical simulation and the experiment is that they run the motor without a load torque, which is not at all comparable to the situation in a vehicle. In a vehicle, the motor is connected via a flexible drivetrain to the vehicle motor, and thus the inertia of the vehicle dominates over the inertia of the rotor, as does the vehicle friction over the friction of the motor. This causes well known issues for the control of traction motors that are not considered here.
The use of an automotive test cycle is therefore not particularly relevant. EUDC is also outdate and no longer used for homologation.
That being said, the experiment seems more appropriate, with typical step responses being tested. If the simulation can be matched to a similar test, I think that would improve the consistency of the paper significantly. If the mentioned points are addressed, the paper should be suitable for publications.
Author Response

(The authors gave the same response as above.)

Reviewer 5 Report
High similarity has been detected @ 36%. suggest to rewrite the sentences and reduced it to below 18%.
The conclusion is badly written, suggest to rewrite the conclusion which can conclude the simulation results and experimental results.
Make some justification between the simulation results and experimental results in term of percentage error, possible constraints, etc.
The overall technical writing style is unsatisfactory. suggest to perform proof reading service. Check the structure and figure placement in the content.
Author Response

(The authors gave the same response as above.)

Round 2
Reviewer 1 Report
The authors have clearly improved the revised version of the manuscript. Here are my suggestion that should be addressed;
1- Check the introduction: "Permanent Magnet Synchronous Motor (SP-PMSM)"
2- Table 1 still has incorrect information. Such as: Controller cost of BLDC can not be very high while PMSM is medium and induction motor is high.
3- I can not accept this paper while table 1 has such big mistakes. Go to original source and check these mistakes.
Author Response

(The authors gave the same response as above.)

Reviewer 4 Report
The paper proposes an adaptive control method for PMSM electric machines, and demonstrates it both in simulation and using experimental results.
The paper is much improved over the first version, and the main criticism has been addressed. The use of language is much better, and I cannot find any obvious mistakes. The contribution is clear and generally well analysed.
I think there are still two limitations that need to be discussed in a bit more detail. The first one is around i_d - which is kept at 0 for this paper. I think that is fine as an initial step, but it will not achieve the wide operating range required for automotive applications. It would be worthwhile to explain this explicitly.
The second limitation is around the torque demand, which is testing in 4.3, but not in 4.1. Usually, for automotive applications, torque and speed profiles are tested together. It is not wrong to separate these, but again it should be explained why.
That being said, the paper does seem suitable for publication now.
Author Response

(The authors gave the same response as above.)

Round 3
Reviewer 1 Report
In my previous report, I advise checking Table 1. Because Table 1 is adapted from the original source [6]. But the original source has a PM column (Separately excited brushed DC) not PMSM. You adapted PM DC motor properties as a PMSM and this is the biggest issue. I advised you to check but you can still not realize what is wrong and deleted the BLDC column. Please clearly check and revise the table. Please adapt the table as given in the source [6].
Author Response

(The authors gave the same response as above.)
